# Periprosthetic Joint Infection (PJI)—Results of One-Stage Revision with Antibiotic-Impregnated Cancellous Allograft Bone—A Retrospective Cohort Study

**DOI:** 10.3390/antibiotics11030310

**Published:** 2022-02-25

**Authors:** Gregor Dersch, Heinz Winkler

**Affiliations:** 1Medical University of Vienna, Spitalgasse 23, 1090 Vienna, Austria; 2Osteitis Center Döbling, Döbling Private Hospital, Heiligenstädter Straße 55-63, 1190 Vienna, Austria

**Keywords:** periprosthetic joint infection, one-stage revision, single-stage, antibiotic-impregnated cancellous allograft bone, impaction grafting, biofilm, microorganism, local antimicrobial therapy, antibiotic carrier, cementless implants

## Abstract

Controversy exists regarding the optimal treatment of periprosthetic joint infection (PJI), considering control of infection, functional results as well as quality of life. Difficulties in treatment derive from the formation of biofilms within a few days after infection. Biofilms are tolerant to systemically applied antibiotics, requiring extreme concentrations for a prolonged period. Minimum biofilm eradicating concentrations (MBEC) are only feasible by the local application of antibiotics. One established approach is the use of allograft bone as a carrier, granting a sustained release of antibiotics in very high concentrations after appropriate impregnation. The purpose of this study was to determine the rate of reinfection after a one-stage revision of infected hip or knee prostheses, using antibiotic-impregnated allograft bone as the carrier and avoiding cement. Between 1 January 2004 and 31 January 2018, 87 patients with PJI, according to MSIS, underwent a one-stage revision with antibiotic-impregnated cancellous allograft bone. An amount of 17 patients had insufficient follow-ups. There were 70 remaining patients (34 male, 36 female) with a mean follow-up of 5.6 years (range 2–15.6) and with a mean age of 68.2 years (range 31.5–86.9). An amount of 38 hips and 11 knees were implanted without any cement; and 21 knees were implanted with moderate cementing at the articular surface with stems always being uncemented. Within 2 years after surgery, 6 out of 70 patients (8.6%, CI 2–15.1) showed reinfection and after more than 2 years, an additional 6 patients showed late-onset infection. Within 2 years after surgery, 11 out of 70 patients (15.7%, CI 7.2–24.2) had an implant failure for any reason (including infection) and after more than 2 years, an additional 7 patients had an implant failure. Using Kaplan-Meier analysis for all 87 patients, the estimated survival for reinfection was 93.9% (CI 88.8–99.1) at 1 year, 89.9% (CI 83.2–96.6) at 2 years and 81.5% (CI 72.1–90.9) at 5 years. The estimated survival for implant failure for any reason was 90.4% (CI 84.1–96.7) at 1 year, 80.9% (CI 72.2–89.7) at 2 years and 71.1% (CI 60.3–81.8) at 5 years. One-stage revision with antibiotic-impregnated cancellous allograft bone grants comparable results regarding infection control as with multiple stages, while shortening rehabilitation, improving quality of life for the patients and reducing costs for the health care system.

## 1. Introduction

Periprosthetic joint infection (PJI) is considered one of the most serious complications in orthopedic surgery. It is associated with prolonged hospitalization and patient immobilization, often leading to emotional [1] and functional morbidity [2,3], as well as remarkable costs for the healthcare system [4,5]. The cumulative incidence has been reported with 1–2% following primary arthroplasty [6,7], although the real number might be higher since the detection of infection is not always feasible [8,9,10,11]. Difficulties to treat PJI derive from microorganisms forming biofilms on necrotic tissue and alloplastic implants [12,13,14], making systemic antibiotics ineffective.

Controversy exists regarding the optimal treatment of PJI. Two-stage exchange is widely regarded as the gold standard method. The reasoning behind the two-stage approach is to apply an enhanced antimicrobial therapy with targeted systemic antibiotics and local antibiotic-loaded cement spacers in between the stages. The fear of reinfection causes most surgeons to wait with implanting a new prosthesis until signs of infection appear to have normalized. This results in multiple stage treatments, lasting several weeks or even months with limited patient mobility and a high rate of spacer-related complications [15,16]. However, the rationale behind two-stage revisions is not evidence-based. There are currently no prospective randomized trials comparing one-stage with two-stage procedures and systematic reviews show comparable results regarding infection control [17,18,19,20,21]. Analysis of pooled longitudinal studies revealed a reinfection rate of 7.6% for one-stage revisions and 8.8% for two-stage revisions of infected total knee arthroplasties [17]. If successful, one-stage revisions provide obvious benefits regarding morbidity, mortality, health care costs [5], functional outcomes [22,23] and patient satisfaction. Since commercially produced spacers increasingly approximate the costs of prosthetic devices, using full prostheses as potentially permanent spacers seems appropriate.

In 2000, the senior author proposed using antibiotic-impregnated cancellous allograft bone to restore lost bone stock and to eradicate microbial pathogens at the same time, making a second procedure unnecessary [24]. Thus far, only a few studies have been performed to evaluate the long-term effectiveness of one-stage revisions using an antibiotic bone compound. First studies showed promising results with reinfection rates of 8% and 3%, respectively [25,26].

The purpose of this study was to determine the rate of reinfection with a minimum follow-up of two years after one-stage revision of PJI according to MSIS [27], using antibiotic-impregnated allograft bone and avoiding cement.

## 2. Materials and Methods

This cohort study was conducted in a secondary and in a tertiary hospital, both located in Austria. Both hospitals specialize in the treatment of PJI, take referrals and use one-stage revisions with antibiotic-impregnated allograft bone as the treatment of choice.

Institutional review board approval was obtained before the initiation of this study (ethics committee of Lower Austria: GS1-EK-4/496-2017; ethics committee of Döbling Private Hospital: 001/2018). Informed consent from the patients for retrospective analysis of medical records was not necessary.

A flowchart illustrates the inclusion, exclusion and results of patients in this study (Figure 1).

The local patient databases were searched for all patients who were treated by the senior author and their medical records were thoroughly examined. This study included all patients who had been operated on by the senior author between 1 January 2004 and 31 January 2018 for a one-stage revision of an infected hip or knee prosthesis, using antibiotic-impregnated cancellous allograft bone.

Periprosthetic joint infections were identified by retrospectively reviewing the medical records and applying the diagnostic criteria of MSIS [27], EBJIS 2018 [28], IDSA (including the clinician’s judgement) [29], IDSA (excluding the clinician’s judgement), ICM 2013 [30], ICM 2018 [31] and the 2018 Definition of Periprosthetic Hip and Knee Infection [32] (Table A1).

For patients who had more than 1 surgery with fitting inclusion criteria during the study period, only the first surgery was included.

In total, there were 124 patients infected according to any PJI definition: 87 patients were infected according to MSIS, 104 according to EBJIS 2018, 117 according to IDSA (including the clinician’s judgement), 88 according to IDSA (excluding the clinician’s judgement), 62 according to ICM 2013, 77 according to ICM 2018 and 77 according to the 2018 Definition of Periprosthetic Hip and Knee Infection. Overlap is demonstrated in Figure 2 and Appendix A.

For further analysis, only infections according to MSIS were considered, as this is the most frequently used protocol in published literature.

Patients with follow-up periods shorter than 2 years after surgery were invited by letter for a follow-up or to answer a questionnaire (Appendix A). Those who failed to respond were contacted by telephone. Patients were excluded if they died within 2 years or had follow-ups shorter than 2 years, despite contact attempts.

Out of 87 patients, 17 patients had an insufficient follow-up: 12 patients had follow-up periods shorter than 2 years, of whom 1 had reinfection and another had reinfection and an amputation. An amount of 4 patients died within 2 years after the surgery; 1 patient with sepsis died 1 day postoperatively due to a recent myocardial infarction and 3 patients died reinfection-free between 15 and 21 months due to lung cancer, pancreatic cancer and stroke, respectively, of whom 2 experienced aseptic implant failure. In one patient, a comminuted pelvic fracture occurred during surgery; this patient underwent a Girdlestone procedure 5 days postoperatively because of mechanical reasons and pain, received a reimplantation surgery after 36 months, was reinfection-free until the last follow-up after 46 months and was excluded due to not having a prosthesis in the postoperative period.

Finally, a total of 70 patients with one-stage revision for infected total joint arthroplasty were included in this study.

The rate of reinfections within 2 years after surgery and the rate of implant failures within 2 years after surgery were calculated.

We also collected other variables, such as age, sex, body mass index (BMI), American Society of Anesthesiologists (ASA) classification score [33], previous surgery history, the existence of sepsis, sinus tracts or purulence, surgery duration, surgery characteristics, culture findings, duration of hospital stay, CRP, ESG and synovial fluid diagnostics.

### 2.1. Definitions

A one-stage prosthesis revision as inclusion criterion was defined as the explantation and implantation of at least an acetabular cup, a femoral component or a tibial component in one procedure. Spacers were counted as a prosthesis at explantation. Arthrodeses were counted as prostheses if they contained intramedullary nails. Arthrodeses fixated by compression plates were not counted as prostheses. Modular component exchanges were not counted as prosthetic revisions.

The diagnostic criteria “culture”, “sinus tract”, and “purulence” were counted if they appeared any time in the period from the index revision until the day after the previous surgery. This time frame was extended until the day after the next previous surgery if the previous surgery did not include the changing of any implant parts, as such, surgeries are thought to be not curative for biofilm infections.

Monomicrobial infection was defined as cultures yielding only one species, regardless of any phenotypical differences expressed by the antibiogram. Polymicrobial infection was defined as cultures yielding different species.

Reinfections were determined by reviewing the medical charts according to the diagnostic criteria of MSIS, irrespective of the occurrence of any reoperation.

Implant failure, for any reason, was defined as a surgical procedure with at least the explantation of an acetabular cup, femoral component or tibial component.

### 2.2. Kaplan–Meier Analysis

To account for patient dropouts, Kaplan–Meier analysis was performed with all 87 PJI revisions according to MSIS in order to estimate the cumulative survival probability. Events of interest were reinfection and implant failure for any reason. In cases of no events of interest, patients were censored at the time of their last follow-up or death. For the endpoint reinfection, patients were also censored at the time of their Girdlestone procedure.

### 2.3. Statistics

The patient data was pseudonymized and collected using Microsoft Excel. Advanced statistical analyses were performed using SPSS 27.0, GraphPad Prism 9 and Microsoft Excel, version 2107. Descriptive statistics are presented in the form of the number of occurrences, percentage and confidence interval or mean, standard deviation and range. Confidence intervals were calculated with the normal approximation method. A confidence level of 95% was applied.

### 2.4. Allograft Bone as Antibiotic Carrier

Preparation and use of the used grafts, following the international standards of the European community, has been extensively described in previous publications [34].

In brief, donors of bone (e.g., femoral heads) are extensively screened by medical history and laboratory tests, including PCR, for detection of viral DNA. Cancellous bone is morselized to pieces in 1 to 10 mm diameters, cleaned and processed by using supercritical carbon dioxide (scCO_2_) [35] with a validated virus-inactivating effect [36]. Lipids and cellular components are completely removed, leaving a pure scaffold of bone matrix. Sterilization is performed by gamma irradiation in a dry state. Since collagen, minerals and osteoinductive proteins remain mainly unaltered, mechanical and biological properties are only slightly modified. The purified matrix is then impregnated with high loads of vancomycin or tobramycin. The standardized impregnation technique provides a uniform concentration of 1 g vancomycin or 0.4 g tobramycin, respectively, in 10 cm^3^ of bone. Due to the proprietary impregnation technique, antibiotics are deposited throughout the whole graft, mainly in the lacunae of the spongy matrix. Finally, the antibiotic bone compound is lyophilized, providing a shelf life of up to 2 years under room temperature. The resulting product is licensed under the trademark OSTEOmycin^TM^. After rehydration, OSTEOmycin^TM^ becomes a moldable mass ready for impaction grafting [37]. After implantation, it elutes vancomycin or tobramycin in a sustained way, providing extremely high local concentrations with a logarithmic decrease over the following weeks [38]. Remodeling of the grafted bone follows the patterns of “creeping substitution” [39].

### 2.5. Treatment Protocol

Access to the infected sites followed the predetermined pathways, with excision of sinus tracts or scars from previous surgeries. In general, the access to the hip joint followed the principles of a transgluteal approach, eventually supplemented by an extension distally for removal of well-fixated stems. In knees, the access was either medially or laterally parapatellar depending on pathways of fistulation or preexisting conditions. In all cases, loose implants were removed and meticulous excision of all cement, granulation, necrotic and infected tissue was performed.

An amount of 5 patients underwent partial revision, retaining femoral stems, tibial stems or acetabular cups, as those implants were fully integrated and without signs of infection. Osteotomy and subsequent cerclage fixation were performed in 7 patients to retrieve firmly connected implants. Plates from previous periprosthetic fractures were explanted in 3 patients. Spacers were explanted in 3 patients. On average, 3.6 tissue samples were sent for culturing and 84% of the patients had implants and/or necrotic bone fragments additionally sent to sonication. Debridement was finalized by extensively using pulsed lavage with saline. After completion of cleaning, gloves, drapes and instruments were changed, and the procedure continued as “aseptic surgery”.

Then the extent of bone defects was evaluated. The remaining defects and medullary cavities were stepwise filled with an average of 92 cm^3^ (range: 40–202 cm^3^) antibiotic-impregnated allograft per surgery, using the “impaction grafting technique” [37]. Solely OSTEOmycin^TM^ was used in 28 patients, solely OSTEOpure^TM^ was used in 14 patients, and both products were used in 28 patients. OSTEOpure^TM^ was manually impregnated with vancomycin or tobramycin before application. To eliminate potentially undetected polymicrobial colonization [11,40], allograft with both vancomycin and tobramycin was used in most cases. Monotherapy with vancomycin was used in cases with strong evidence of monomicrobial Gram-positive infection, such as the acute onset of symptoms and typical clinical appearance (fever, pus).

All prosthetic implants were anchored following the principles of press-fit fixation. Fractures were stabilized with intramedullary stems and cerclage bands. In all 38 hip revisions, no cement was used. Out of the 32 knee revisions, in 21 (66%), cement was used in moderate amounts at articular surfaces. All stems were implanted cement-free. Fixation was intraoperatively qualified as stable in all cases.

Wounds were drained and closed. No plastic surgery flaps or skin grafts were necessary for the index revisions for complete soft tissue coverage. Operating time ranged from 2 to 6 h. Rehabilitation was commenced as per a routine revision operation. Perioperative systemic antibiotic treatment was adjusted to the results of preoperative and intraoperative cultures. According to our protocol, 2 weeks of intravenous followed by 4 weeks of oral antibiotic therapy was administered, however, some cases required premature termination due to an intolerance to prescribed antibiotics. In case of culture-negative infection, a second-generation cephalosporin was generally given until results of intraoperative cultures became available, especially respecting the results of sonication. Medical complications (venous thromboembolism, wound complications, urinary tract infection, etc.) and blood loss were in the range of aseptic revisions.

## 3. Results

There were 70 patients who were treated by the senior author with one-stage revisions for PJI according to MSIS, using antibiotic-impregnated cancellous allograft bone. The mean follow-up was 5.6 years (range 2–15.6). The patient characteristics are shown in Table 1.

The microbiological results of intraoperative cultures, sonications, preoperative aspirations and deep sinus tract cultures for the index revisions can be found in Table 2. More detailed microbiological results for the index revisions and reinfections within 2 years are in Appendix A.

Within 2 years after surgery, 6 out of 70 patients (8.6%, CI 2–15.1) showed reinfection and after more than 2 years, an additional 6 patients showed late-onset infection.

Out of the 6 patients with reinfections within 2 years (3 hips, 3 knees), 2 developed postoperative fistulas and did not receive further surgery, 1 patient received irrigation, debridement and removal of infected heterotopic ossification and stayed infection-free until his last follow-up 52 months later, 1 patient received irrigation and debridement and later a partial re-revision, 1 patient received multiple re-revisions and 1 patient had a modular component exchange and later an amputation. An amount of 3 out of the 6 reinfections were monomicrobial infections with different species than in the index revisions. A further 2 reinfections each occurred with a previously identified organism (*Staphylococcus epidermidis*) plus a newly occurring organism.

Within 2 years after surgery, 11 out of 70 patients (15.7%, CI 7.2–24.2) had an implant failure for any reason and after more than 2 years, an additional 7 patients had an implant failure.

The causes for the 11 implant failures within 2 years (5 hips, 6 knees) were reinfection (n = 3), periprosthetic acetabular fracture (n = 2), aseptic loosening (n = 2), periprosthetic femoral fracture (n = 1), recurring dislocations (n = 1) and unknown causes (n = 2).

Two Kaplan–Meier curves of 87 patients, including patients with insufficient follow-ups, illustrate the cumulative survival for reinfection and implant failure for any reason in Figure 3.

The estimated survival for reinfection was 93.9% (CI 88.8–99.1) at 1 year, 89.9% (CI 83.2–96.6) at 2 years and 81.5% (CI 72.1–90.9) at 5 years.

The estimated survival for implant failure for any reason was 90.4% (CI 84.1–96.7) at 1 year, 80.9% (CI 72.2–89.7) at 2 years and 71.1% (CI 60.3–81.8) at 5 years.

## 4. Discussions

In the treatment of chronic PJI, complete removal of infected implants and radical debridement of any infected or avascular tissue are considered prerequisites for successful revision. However, even after the most careful cleaning, small fragments of biofilm may be displaced to new habitats in niches of the debrided site, potentially leading to reinfection. Therefore, any local antibiotic delivery system should at least reach minimum biofilm elimination concentrations while simultaneously providing filler material for dead space. In vitro studies showed that minimum inhibitory concentrations are around 2 mg/L of vancomycin for planktonic cocci, but minimum biofilm elimination concentrations of *Staphylococcus aureus* have been observed to reach 2000–8000 mg/L when applied for 3–5 days [41,42], and 200 mg/L when applied for 28 days [43]; concentrations impossible to achieve with systemic antibiotic therapy [44,45,46].

Antibiotic-loaded cement shows strongly varying antibiotic release kinetics throughout the literature. Conventional mixing methods consist of adding 0.5–2 g antibiotics to 40 g cement, achieving in vitro concentrations of 70–800 mg/L on the first day and 6–25 mg/L on day 5 [47,48]. However, 80–99% of antibiotics remain trapped inside the cement, indicating that antibiotic elution is primarily a surface phenomenon [48,49,50,51,52,53,54]. Higher concentrations have been achieved by adding more antibiotics [55,56] or by changing the mixing technique [53] but increasing the cement’s porosity comes at the cost of worsened mechanical properties [52,57,58,59,60,61]. It has been reported that 50% of antibiotic-loaded spacers and 90% of antibiotic-loaded beads are covered with biofilms at removal, further indicating insufficient protection against bacteria [62,63,64].

When cancellous allograft bone is being cleaned from all fatty bone marrow, soft tissue and cells, it acts as a natural carrier releasing all antibiotics [34]. An amount of 50 cm^3^ (ψ 9 g) highly purified cancellous allograft bone can be loaded with 5 g vancomycin. In vitro, concentrations of 21,000 mg/L on the first day and 100 mg/L on day 7 can be achieved [38], and mature *Staphylococcus aureus* biofilms in vitro can be eradicated [42]. Despite the very high local vancomycin concentrations, in vivo studies showed no adverse effects [25,65], confirming the low cytotoxic properties [66] and poor penetration to the vascular system [44,65].

Bone cement connects strongly with the underlying bone, which makes any potential further revision laborious and causes large bone defects at removal. This is especially critical in high-risk patients with potentially recurrent infections. In contrast, an antibiotic bone compound in uncemented revisions has a good chance of being integrated into osseous defects, restoring bone stock with each consecutive surgery. The allograft bone appears to be capable of limited weight-bearing from the beginning and shows rapid incorporation (Figure 4) [67].

In our cohort study of one-stage revisions for PJI using antibiotic-impregnated cancellous allograft bone, 6 out of 70 patients (8.6%, CI 2–15.1) had reinfection within 2 years and the estimated reinfection rate at 5 years according to Kaplan–Meier analysis was 18.5% (CI 9.1–27.9).

These are comparable results to two-stage procedures but without the obvious disadvantages of two or more procedures and the period of disability in between them [17,18,19,20,21].

Many authors believe strict inclusion criteria to be necessary for one-stage revision: Sinus tracts, no preoperatively identified organisms, difficult-to-treat organisms, history of previous surgeries for PJI or large bone defects are usually contraindications for direct exchange. However, we demonstrated a treatment protocol effective even in high-risk patients: 30% of our patients had a sinus tract, 50% had a previously failed surgical treatment for PJI, the average number of previous surgeries was 3.2, and on average, 92 cm^3^ allograft was used.

Our 15.7% (CI 7.2–24.2) implant failure rate within 2 years and 28.9% (CI 18.2–39.7) estimated implant failure rate at 5 years, according to Kaplan–Meier analysis was comparable to the number of complications occurring with multiple stage procedures [3,68,69]. An amount of 3 out of 70 (4.3%) patients suffered from periprosthetic fractures within 2 years after the surgery, of whom all 3 had uncemented hip implants. One possible explanation might be that impaction grafting or press-fit fixation places increased stress on patients’ bone structures, especially in high-risk patients with large bone defects, increasing the risk of fractures.

An amount of 2 patients with persistent draining sinuses were in poor general health, had comparatively little discomfort and thus were managed without further surgery. Sometimes it may be more advantageous to accept a recurrence of infection and provide fast rehabilitation than to risk loss of function, limb or life [70,71].

An amount of 38 out of 70 (54.3%) patients had a polymicrobial infection, which is in stark contrast to the usual 1–10% polymicrobial infections reported by other studies about PJI. There was a remarkable percentage of methicillin-resistant Gram-positive bacteria present, especially in coagulase-negative staphylococci. However, since all of them were susceptible to vancomycin, it did not influence our treatment protocol and only directed our postoperative systemic therapy. This was similar to the occurring Gram-negative bacteria, which all were susceptible to tobramycin.

There were several differences between our study and other similar studies, potentially reducing the comparability. First, our study counted reinfections irrespective of reoperations. Many studies about one-stage or two-stage revisions for PJI define failure as revision due to PJI [72,73,74,75,76] or reoperation due to PJI [77,78,79,80,81], which might lead to an underestimation of the true rate of reinfection.

Furthermore, most studies about two-stage revisions exclude all patients who received the first procedure but failed to undergo the second procedure [75,76,77,78,79,80,82,83,84,85,86], thereby introducing survivorship bias. An amount of 7–29% of patients never undergo reimplantation because they are considered to be too unfit for further surgical reconstruction, decline, die or undergo amputation [3,68,69,73,76,80,81,84,85,86,87]. Studies including patients who fail to receive the second procedure, report failure rates of more than 30% [68,73]. In light of these findings, the high success rates of two-staged revisions may need to be re-examined.

Moreso, most two-stage revision studies do not consider repeated irrigation and debridement or repeated spacer exchanges because of persisting infection in between the stages as a failure.

We observed vast differences in the PJI-classification of patients depending on the diagnostic scoring systems. These findings were in accordance with Renz et al. [28] and Huard et al. [88], but contradictory to the results of Guan et al. and Melendez et al. [89,90]. More patients were diagnosed as positive according to EBJIS 2018 or IDSA, because merely one positive culture in specific circumstances—purulence or positive histology—was already enough for a positive diagnosis. The largest number of patients being diagnosed as infected according to the IDSA diagnostic criteria (including the clinician’s judgement) might be explained by the retrospectively missing medical records of referring hospitals to which the originally diagnosing clinicians had access. In those cases, the extensive prior treatment with antibiotics might have limited culture retrieval in our treatment centers. Considerably fewer patients had an infection according to ICM 2013, ICM 2018 or the 2018 Definition of Periprosthetic Hip and Knee Infection, since those diagnostic systems require the presence of the same organism or phenotypically identical organisms. Out of 80 patients with ≥2 positive cultures, 18 (22.5%) had exclusively different species in their cultures and 30 (37.5%) had no documented phenotypically identical species in their cultures. Out of 51 patients with ≥3 positive cultures, 2 (3.9%) had exclusively different species in their cultures and 8 (15.7%) had no documented phenotypically identical species in their cultures. These patients would be considered not infected according to the major culture criteria of ICM 2013, ICM 2018 or the 2018 Definition of Periprosthetic Hip and Knee Infection.

The strong lack of consensus in the selection of patients and in the definition of treatment failure might explain the vast differences in reported surgical outcomes across the literature, ranging from 0–40% reinfection rates [18,19]. Tan et al. reported strongly varying success rates of 54.2–96.8% in the same patient cohort, depending on the definition of success being used [87]. An international consensus meeting attempted to standardize the definition of success after PJI treatment using the Delphi method [91]. However, Tan et al. reported that the Delphi consensus definition could not be assessed in 19.6% of PJI cases as reimplantation for the planned two-stage exchange never occurred [87].

There are several limitations to our retrospective cohort study. It has a lower level of evidence than a randomized controlled trial. This is due to the missing random assignment into different groups, which precludes the ability to control for unknown or unmeasured confounding variables. Moreso, there might be a selection bias if the patient group of this study is not comparable with the groups of other studies. For example, many patients were referred to our treatment centers because of failed previous surgical interventions to treat PJI. A large number of patients were excluded due to insufficient follow-up. To mitigate this bias, Kaplan–Meier analysis was performed and included all patients with insufficient follow-ups, which showed similar estimated survival results. Finally, erythrocyte sedimentation rate and synovial fluid parameters have not been documented in the majority of medical records and therefore could not be used for the PJI scoring systems in those cases.

However, our study also has some strengths: At the present time, to the best of our knowledge, it is the largest cohort study regarding one-stage revision using antibiotic-impregnated allograft bone for the treatment of PJI. In addition, patients with a variety of medical histories, organisms and with varying bone quality were treated by one surgeon with a consistent treatment protocol. Other advantages are the long follow-ups with an average of 5.6 years and the application of enhanced culture methods (sonication) in 84% of cases.

## 5. Conclusions

One-stage revision with antibiotic-impregnated cancellous allograft bone grants comparable results regarding an infection control as with multiple stages, while shortening rehabilitation, improving the quality of life for patients and reducing health care system costs.

Standardized, validated, and applicable definitions of PJI and treatment success are needed to increase the comparability between studies.

Randomized controlled trials should further determine superior results of either one-stage or two-stage revisions.

## Figures and Tables

**Figure 1 antibiotics-11-00310-f001:**
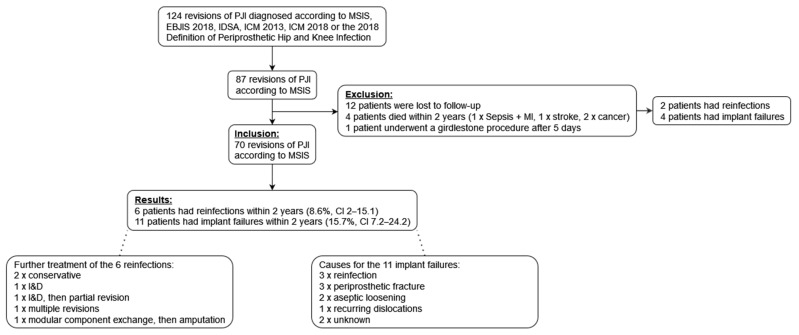
Flowchart illustrating the inclusion, exclusion and results of patients in this study. Abbreviations: MI, myocardial infarction; I&D, irrigation and debridement.

**Figure 2 antibiotics-11-00310-f002:**
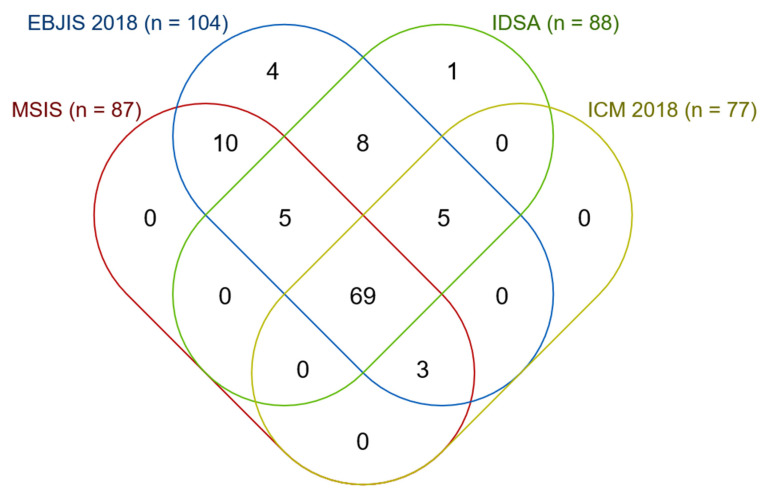
Venn diagram (n = 124) demonstrating the overlap of patients diagnosed according to MSIS, EBJIS 2018, IDSA and ICM 2018.

**Figure 3 antibiotics-11-00310-f003:**
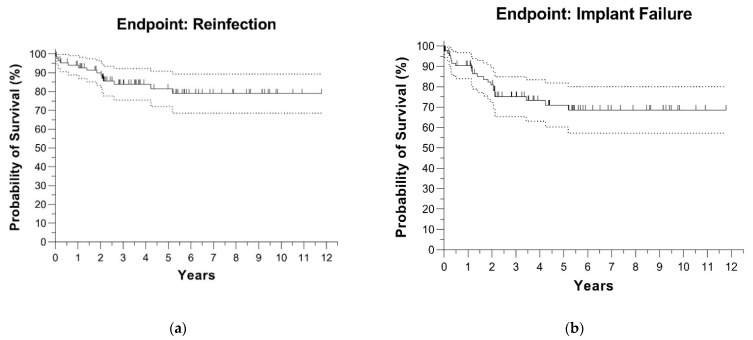
Kaplan–Meier curves with (**a**) reinfection and (**b**) implant failure for any reason as events of interest. All 87 patients with PJI according to MSIS are included. The dotted lines are the confidence intervals per x-value, the vertical stripes are censored events.

**Figure 4 antibiotics-11-00310-f004:**
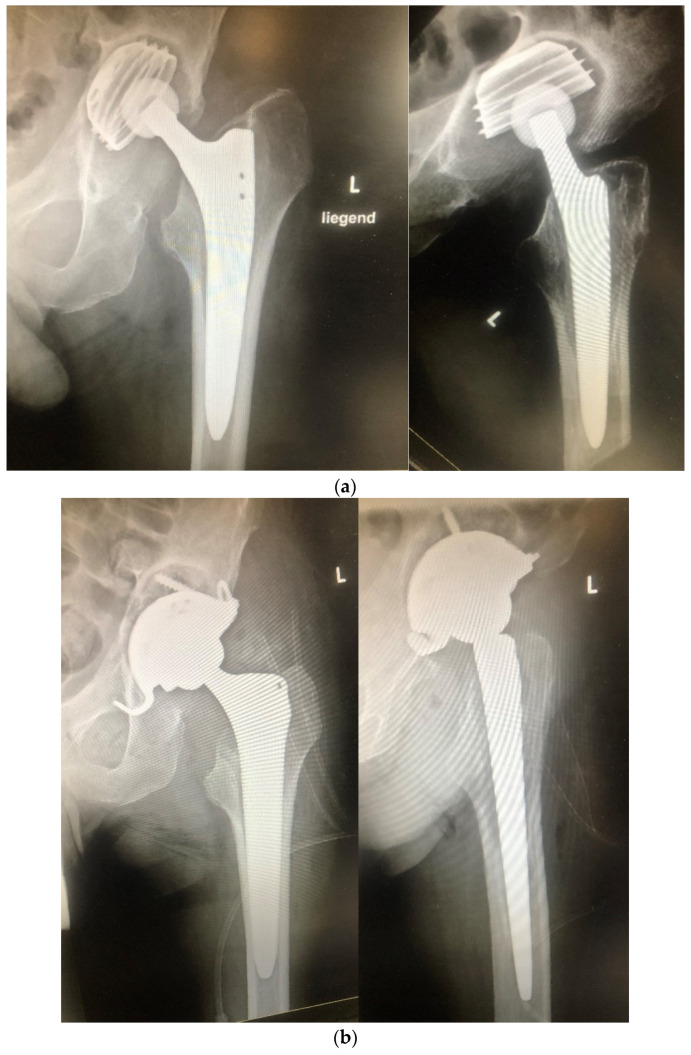
Radiographs of a 66-year-old male who sustained a femoral neck fracture treated with uncemented total hip arthroplasty. Postoperatively, he complained of unspecific pain with only slightly elevated infection markers. (**a**) Three years later, loosening of the acetabular component was diagnosed with marked osseous defect periacetabular and signs of osteolysis around the proximal part of the stem. (**b**) One-stage exchange with uncemented components. The defects were filled with antibiotic-impregnated bone (OSTEOmycin^TM^ V). Sonication of explanted material revealed growth of two strains of methicillin-resistant *S. epidermidis* and *Propionibacterium* sp. The hospital stay was one week, with cefuroxime intravenously, followed by six weeks of amoxicillin/clavulanic acid and rifampicin orally. (**c**) Six months postoperatively, the patient is pain-free with no sign of infection and unlimited mobility. There is partial remodeling of the allograft. Figures taken from previous publication in EOR Winkler (2017) [67]. (**d**) 6,5 years postoperatively. Unlimited weight-bearing, no sign of infection. No change of position of implants. The allograft is completely incorporated with seamless ingrowth of cup and stem.

**Table 1 antibiotics-11-00310-t001:** Patient characteristics (n = 70).

Characteristics	Values
**Preoperative**	
Age (years)	68.2 ± 12.3 (31.5–86.9)
Sex	
Male	34 (48.6%)
Female	36 (51.4%)
BMI (kg/m^2^) ^a)^	29.3 ± 6 (19.5–46.1)
ASA score ≥ 3 ^a)^	20 (52.6%)
History of surgery for PJI on respective joint	35 (50%)
Previous surgeries on respective joint	3.2 ± 2.8 (1–13)
≥8 previous surgeries on respective joint	7 (10%)
Sepsis as surgery indication	4 (5.7%)
Sinus tract	21 (30%)
Purulence	37 (52.9%)
Intraoperative	
Surgery duration (minutes) ^a)^	187.8 ± 47.8 (114–305)
Surgery site	
Knee joint	32 (45.7%)
Hip Joint	38 (54.3%)
Spacer explantation	3 (4.3%)
Osteosynthesis plate removal	3 (4.3%)
Volume antibiotic-impregnated allograft bone (cm^3^)	92.2 ± 27.8 (40–202)
Arthrodesis Implantation	1 (1.4%)
Cementless prosthesis implantation	49 (70%)
Application of intramedullary cement	0 (0%)
**Postoperative**	
Hospital stay after surgery (days)	16.8 days ± 8.7 (7–47)

Data are mean ± standard deviation (range) or number (%) of episodes. ^a)^ Due to retrospectively missing documentation, these values were calculated only from 38 patients from one treatment center.

**Table 2 antibiotics-11-00310-t002:** Identified microbial pathogens in 70 patients.

Microorganisms	Monomicrobial Infections (n = 32), No. of Infected Patients	Polymicrobial Infections (n = 38), No. of Infected Patients
**Fungus:**		
*Candida albicans*	1	
**Gram-negative bacteria:**		
*Pseudomonas aeruginosa*	2	
*Escherichia coli*	1	
*Klebsiella pneumoniae*		1
*Proteus vulgaris*		1
*Fusobacterium* sp.		1
*Ralstonia picketti*		1
**Gram-positive bacteria:**		
*Micrococcus* sp.		4
*Brevibacterium casei*		1
*Propionibacterium* sp.	3	9
*Corynebacterium* sp.		5
*Bacillus* sp.		2
*Staphylococcus aureus*	1	6
*Staphylococcus epidermidis*	16	24
*Staphylococcus warneri*		3
*Staphylococcus capitis*	1	3
*Staphylococcus hominis*		3
*Staphylococcus lugdunensis*	1	1
*Staphylococcus caprae*		1
*Staphylococcus haemolyticus*		1
“Mixed growth of *Staphylococcus* spp.”		2
Coagulase-negative *Staphylococcus* sp., unspecified	2	7
*Gemella morbillorum*		1
*Streptococcus* sp.	3	6
*Enterococcus* sp.	1	5
*Aerococcus viridans*		1
*Finegoldia magna*		2
*Peptostreptococcus* sp.		2
*Peptococcus* sp.		1
*Anaerococcus prevotii*		1
unspecified Species		2
Culture-negative	none	none

## Data Availability

The pseudonymized data presented in this study are available from the corresponding authors upon reasonable request. The original medical records in Döbling Private Hospital, Osteitis Center Döbling and Landesklinikum Korneuburg are not publicly available due to data privacy regulations.

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
