# Peer review of "Periprosthetic Joint Infection (PJI)—Results of One-Stage Revision with Antibiotic-Impregnated Cancellous Allograft Bone—A Retrospective Cohort Study"

_antibiotics, 2022, doi:10.3390/antibiotics11030310_

Round 1

Reviewer 1 Report

Thank you for the opportunity to read this interesting manuscript. The authors reported the results of one-stage PJI revision with antibiotic-impregnated cancellous allograft bone. The number of patients is relatively large, and the results are promising. However, there are a few concerns that should be revised.

The authors should describe surgical time, intraoperative blood loss and transfusion, surgical approach(hip), and postoperative complications (transfusion rate, venous thromboembolism, wound complications, urinary tract infection, and other medical complications). Further, the number of administration days of intravenous and oral antibiotics should be reported.

Line 226 and 283

There were 6 patients who had reinfection within 2 years. And there were 3 reinfection patients classified to implant failure, according to Figure 1. Were the latter 3 patients included in the former 6 reinfection patients? If so, that should be clearly described in the text and figure.

Table 2

The authors show the pathogenic microorganisms in Table 2. Were all organism sensitive for normal antibiotics? Was no MRSA identified? If there were antibiotic-resistant organisms, that should be noted.

Line 25

I think that 10 (14.3%, CI 6.1-22.5) should be revised to 11 (15.7%, CI 7.2-24.2).

Author Response

Reviewer 1

Thank you for the opportunity to read this interesting manuscript. The authors reported the results of one-stage PJI revision with antibiotic-impregnated cancellous allograft bone. The number of patients is relatively large, and the results are promising. However, there are a few concerns that should be revised.
 We thank the reviewer for his encouraging words and appreciate the suggestions for improving the paper.

The authors should describe surgical time, intraoperative blood loss and transfusion, surgical approach(hip), and postoperative complications (transfusion rate, venous thromboembolism, wound complications, urinary tract infection, and other medical complications). Further, the number of administration days of intravenous and oral antibiotics should be reported.

We agree there are a lot of features worth being covered, as mentioned by the reviewer. However, the scope of this work has focused on the final results regarding infection control and mechanical outcome. Including all interesting details into the protocol would have been out of the scope of this paper although certainly of interest for further stratification of the results. We have included respective general remarks, as available, in the manuscript and hope further research might provide more detailed information on possible effects of medical conditions on the final outcome

Line 226 and 283

There were 6 patients who had reinfection within 2 years. And there were 3 reinfection patients classified to implant failure, according to Figure 1. Were the latter 3 patients included in the former 6 reinfection patients? If so, that should be clearly described in the text and figure.

Yes, the 3 reinfected and revised patients were included in both. We made some clarification in the text.

Table 2

The authors show the pathogenic microorganisms in Table 2. Were all organism sensitive for normal antibiotics? Was no MRSA identified? If there were antibiotic-resistant organisms, that should be noted.

We did not record resistance patterns of the pathogens. Indeed there was a remarkable percentage of methicillin resistant Grampositives present, especially in CNS. However, since all of them were susceptible to Vancomycin it did not influence our treatment protocol and only directed our postoperative systemic therapy. Similar with Gramnegatives, which all were susceptible to Tobramycin. We have added a respective remark in the text. There were no cases with MRSA showing a recurrence.

Line 25

I think that 10 (14.3%, CI 6.1-22.5) should be revised to 11 (15.7%, CI 7.2-24.2).

Thank you for the note. We have corrected the respective numbers.

Reviewer 2 Report

Periprosthetic joint infections are a severe and difficult to treat complication in orthopedic surgery. While there is still controversy regarding the best treatment approach, the authors evaluated retrospectively the rate of reinfection after 2 years in a group of patients treated with one-stage revision operation using antibiotic-impregnated allograft bone. Comparable infection control to other treatment approaches was found.

General comments:

The authors provide data on a large group of patients with a long follow up. The manuscript is well written. Methods are described comprehensibly. Different diagnostic criteria for periprosthetic joint infection are applied for identifying patients leading to an interesting evaluation of the overlap of patients by the various criteria. While this situation is adequately discussed and adequate conclusions are drawn, a short explanation why the authors considered only infections according to MSIS for further analysis, would be helpful.

It would be helpful, if the authors could give some more patient characteristics regarding risk factors for wound infection (e.g. obesity – weight or body mass index; diabetes mellitus; rheumatoid arthritis, malignancies: steroid administration; days of wound secretion and drainage).

Was there any additional systemic or local administration of antibiotics? How long?

Specific comments:

Materials and methods:

Lines 90-91 “…..fitting the inclusion criteria……” – what are the inclusion criteria explicitly?

Lines 102-105: Patients with follow-up periods less than 2 years were contacted in several ways and then excluded? How did the authors receive the data of the patients that were included and had a follow-up between 2 and 15 years (according to page 6 line 212)?

Line 214: titel of table 1: Patient characteristics (n = 70) is sufficient;

Lines 97 and 103: the mentioned supplementary figures were not available for review

Author Response

Reviewer 2

Periprosthetic joint infections are a severe and difficult to treat complication in orthopedic surgery. While there is still controversy regarding the best treatment approach, the authors evaluated retrospectively the rate of reinfection after 2 years in a group of patients treated with one-stage revision operation using antibiotic-impregnated allograft bone. Comparable infection control to other treatment approaches was found.

General comments:

The authors provide data on a large group of patients with a long follow up. The manuscript is well written. Methods are described comprehensibly. Different diagnostic criteria for periprosthetic joint infection are applied for identifying patients leading to an interesting evaluation of the overlap of patients by the various criteria. While this situation is adequately discussed and adequate conclusions are drawn, a short explanation why the authors considered only infections according to MSIS for further analysis, would be helpful.

We thank the reviewer for his encouraging statement. We have indicated, unlike to other publications, that there exist several classifications for PJI, all showing advantages and disadvantages. As a choice needed to be done we decided to make MSIS the basis of our evaluation because this one is most widely used in the existing literature, such making our results more comparable with existing publications on the topic.

It would be helpful, if the authors could give some more patient characteristics regarding risk factors for wound infection (e.g. obesity – weight or body mass index; diabetes mellitus; rheumatoid arthritis, malignancies: steroid administration; days of wound secretion and drainage).

We agree with the reviewer that all the mentioned patient characteristics would be interesting to be investigated. However, the scope of this work has focused on the final long term results regarding infection control and mechanical outcome. Including all interesting details into the protocol would have been out of the scope of this paper although certainly of interest for further stratification of the results. We have included respective general remarks, as available, in the manuscript and hope further research might provide more detailed information on possible effects of medical conditions on the final outcome

Was there any additional systemic or local administration of antibiotics? How long?

Local administration was limited to Vancomycin or Tobramycin or a combination of both. No other antibiotics were placed locally. Systemic therapy always followed the resistance patterns of found pathogens, based on preoperative cultures and eventually modified after receiving results of intraoperative cultures. As all explanted devices were sonicated we were able to identify the causing pathogen in all cases, including some that revealed no growth preoperatively. According to our protocol systemic antibiotic therapy consisted of two weeks of intravenous followed by 4 weeks of oral administration; however, this regimen was shortened in some cases where the antibiotics produced unacceptable side effects. There was no case with antibiotic treatment longer than 6 weeeks.

Specific comments:

Materials and methods:

Lines 90-91 “…..fitting the inclusion criteria……” – what are the inclusion criteria explicitly?

For our study all patients were included with a PJI according to MSIS and minimum 2yr follow-up, irrespective of underlying conditions. We have added a more precise description in the text.

Lines 102-105: Patients with follow-up periods less than 2 years were contacted in several ways and then excluded? How did the authors receive the data of the patients that were included and had a follow-up between 2 and 15 years (according to page 6 line 212)?

We agree the sentence could have been misleading and inserted a correction in the text.

Line 214: titel of table 1: Patient characteristics (n = 70) is sufficient;

We have corrected as suggested

Lines 97 and 103: the mentioned supplementary figures were not available for review

We apologize for having forgotten to send the respective supplementary. It is added to our answer.
